# Effects of a Novel Amino Acid Formula on Nutritional and Metabolic Status, Anemia and Myocardial Function in Thrice-Weekly Hemodialysis Patients: Results of a Six-Month Randomized Double-Blind Placebo-Controlled Pilot Study

**DOI:** 10.3390/nu14173492

**Published:** 2022-08-25

**Authors:** Stefano Murtas, Roberto Aquilani, Gianmarco Fiori, Roberto Maestri, Paolo Iadarola, Cristina Graccione, Rita Contu, Maria Luisa Deiana, Fabrizio Macis, Romina Secci, Antonella Serra, Mariella Cadeddu, Maura D’Amato, Paola Putzu, Mirella Marongiu, Piergiorgio Bolasco

**Affiliations:** 1Nephrology Department, ASL of Cagliari, 09100 Cagliari, Italy; 2Department of Biology and Biotechnology “Lazzaro Spallanzani”, University of Pavia, 27100 Pavia, Italy; 3Cardiology Service—Quartu-Parteolla Health District, ASL of Cagliari, Quartu Sant’Elena, 09045 Cagliari, Italy; 4Department of Biomedical Engineering, Scientific Institute of Montescano, IRCCS, ICS Maugeri S.p.A SB, 27100 Pavia, Italy; 5Department of Molecular, Medicine University of Pavia, 27100 Pavia, Italy; 6Chronic Kidney Disease Treatment Conservative Study Group of the Italian Society of Nephrology, 00185 Rome, Italy

**Keywords:** hydration, hemodialysis, amino acid replacement, metabolic and nutritional disorders

## Abstract

(1) Background: Chronic Kidney Disease (CKD) induces metabolic derangement of amino acid (AA) kinetics, eliciting severe damage to the protein anabolism. This damage is further intensified by a significant loss of AAs through hemodialysis (HD), affecting all tissues with a high metabolic turnover, such as the myocardium and body muscle mass. (2) Aim: to illustrate the effects of a novel AA mixture in boosting mitochondrial energy production. (3) Methods: A strict selection of 164 dialysis patients was carried out, allowing us to finally identify 22 compliant patients who had not used any form of supplements over the previous year. The study design envisaged a 6-month randomized, double-blind trial for the comparison of two groups of hemodialysis patients: eleven patients (67.2 ± 9.5 years) received the novel AA mix (TRG), whilst the other eleven (68.2 ± 10.5 years) were given a placebo mix that was indistinguishable from the treatment mix (PLG). (4) Results: Despite the 6-month observation period, the following were observed: maintenance of target hemoglobin values with a reduced need for erythropoiesis-stimulating agents in TRG > 36% compared to PLG (*p* < 0.02), improved phase angle (PhA) accompanied by an increase in muscle mass solely in the TRG group (*p* < 0.05), improved Left Ventricular Ejection Fraction (LVEF > 67%) in the TRG versus PLG group (*p* < 0.05) with early but marked signs of improved diastolic function. Increased sensitivity to insulin with greater control of glycemic levels in TRG versus PLG (*p* = 0.016). (5) Conclusions: the new AA mix seemed to be effective, showing a positive result on nutritional metabolism and cardiac performance, stable hemoglobin levels with the need for lower doses of erythropoietin (EPO), insulin increased cell sensitivity, better muscle metabolism with less loss of mass.

## 1. Introduction

Uremia-induced negative effects are evident in the terminal stages of chronic kidney disease (CKD4-5: Glomerular Filtration Rate: 29 and <15 mL/min), resulting in a significantly increased risk of cardiovascular comorbidity-mortality and nutritional deficits [1]. The complications produced by chronic uremia are further exacerbated by hemodialysis, which raises the risk of morbidity and mortality associated with a wide spectrum of diseases arising from the state of diffuse chronic inflammation and oxidation [2], resulting in a worsening of anorexia, dyslipidemia [3] and microbiological derangement of gut microbiota [4]. Moreover, frequent reports of anorexia in these patients inevitably herald the onset of Protein Energy Wasting (PEW) [5,6] over a relatively short-to-medium time span. Although largely underestimated in recent years, renewed interest has recently been focused on the unexpected, substantial loss of amino acids (AAs) in dialysate. AA loss has long been an acknowledged [7,8], although frequently underestimated, phenomenon, and the severe metabolic consequences caused by such heavy AA losses in the medium-long term have been overlooked. Several recent literature reports have reiterated how AA losses may be as high as ≥700–800 g/year [9,10]. Diffusive techniques are known to produce significant intradialytic AA losses, although following the relatively recent introduction of combined diffusive-convective methods, including hemodiafiltration, the applied ultrafiltration-convective forces intensify the exchange of plasma water with reinfusion liquid [11] and a higher loss of AAs, resulting in a further alteration of the AA metabolism, the fate of which is linked to the progression of CKD and frequency of hemodialysis. The outcome and metabolic changes of individual AAs in CKD and during hemodialysis remain to be clarified, particularly in subjects aged > 65–70 years, in whom an already decreased sensitivity to insulin tends to hamper intra-cellular access and the use of AAs [12], resulting in inhibition of AA uptake in protein synthesis and onset of protein catabolism, particularly in muscle mass, of dialysis patients [13]. Future research should focus on developing tailored amino acid mixtures for all stages of CKD to compensate for the loss of different classes of AAs. The use of amino acid mixtures is fundamental to counteract dialysis-related AA loss, whilst also providing added benefits for organs and systems with a high metabolic turnover, including the cardiovascular system and muscle mass, and correction of anemia stemming from reduced erythropoietin (EPO) synthesis, particularly in advanced CKD and hemodialysis patients [14], with the aim of reducing morbidity and mortality and improving quality of life.

The aim of our study, therefore, was to compare two groups of patients subjected to thrice-weekly hemodialysis following administration of a novel micronutrient-enriched amino acid mixture containing metabolic and mitochondrial accelerators. The HD patient population was made up of a cohort treated with the new mixture versus a control group of HD patients treated with a placebo.

## 2. Materials and Methods

The initial hemodialysis patient population contacted comprised 164 patients from five different dialysis centers. Strict exclusion criteria were established, with the main focus directed at precluding recruitment of patients taking oral or parenteral supplements, multi-vitamins and/or other protein/amino acid products to prevent interference in the study outcome, thus resulting in the identification of only 26 eligible patients. Additional exclusion criteria included: presence of residual kidney function (RKF) and diuresis >200 mL/day, patients treated with a twice-once weekly HD schedule, and presence of diseases characterized by derangement of metabolic steady state, i.e., acute or chronic inflammatory diseases, type I or II diabetes, malignant tumors, autoimmune diseases, treatment with steroids and/or immunosuppressant drugs, chronic pulmonary disease, malnutrition, cardiomyopathy with heart failure, and liver disease. Four patients were excluded from the study during the first month due to limited diet adherence in taking the amino acid mixture. The remaining 22 patients were aged 67.9 ± 9.6 years, with a dialysis vintage of 95.9 ± 8.9 months. The study design envisaged double-blind centralized randomization and comparison of two groups, the first comprising 11 patients aged 67.2 ± 9.5 years with a dialysis vintage of 123.7 ± 121.7 months (7 females, 4 males) who were treated with the new amino acid mixture (TRG) and a second group of 11 patients aged 68.2 ± 10.5 years with a dialysis vintage of 51.5 ± 39.5 months (6 females, 5 males) who received placebo (PLG). Sachets of Amino-Ther-PRO ^®^ amino acid mixture (Professional Dietetics S.r.l.—Milan, Italy) contained: L-Leucin: 1200 mg, L-lysine: 900 mg, L-Threonine: 700 mg, L-Isoleucine: 600 mg, L-Valine: 600 mg, L-Cysteine: 150 mg, L-Histidine: 150 mg, L-Phenylalanine: 100 mg, L-Methionine: 50 mg, L-Tryptophan: 50 mg, Vitamin B6: 0.85 mg, Vitamin B1: 0.70 mg, citric acid: 409 mg, malic acid: 102.5 mg, succinic acid: 102.5 mg, beta-carotene 1%: 10 mg, energy: 33 Kcal. A weekly total of 31.5 g AAs was given. The novel AA mixture was administered on a two-monthly basis to achieve a total AA delivery of >700 g to counterbalance mean hemodialytic AA losses amounting to 750 g/year [9]. This regimen may also boost treatment compliance. To prevent drag of catabolic effects from the hemodialysis session, contents of the sachet were taken 4–5 h after the end of HD morning sessions and one of the HD evening sessions on non-dialysis days. Sachets of placebo were taken according to the same schedule and contained: maltodextrin: 5343.2 mg; tropical aroma: 180 mg; sucrose ester: 180 mg; hydroxypropyl cellulose: 63.4 mg; sucralose: 28 mg; silica dioxide: 25 mg; acesulfame k: 23 mg; Vitamin B6: 1.0 mg; Vitamin B1 0.9 mg; beta-carotene 1%: 10 mg. The treatment mixture also contained a series of energy-boosting compounds, including succinic acid which, in the presence of acetyl-CoA, stimulates the activation of the succinate dehydrogenase enzyme, resulting in increased mitochondrial membrane permeability, increased oxygen absorption and a higher production of ATP. Malic acid, which activates a malate dehydrogenase reaction promoting ingress of cytosolic NADH into the otherwise impermeable mitochondria, was included to enhance mitochondrial activity and absorption of O2 [15,16], together with citric acid to boost mitochondrial turnover. Other actions described at cell level include the facilitating of insulin synthesis and post-ischemic recovery of myocytes [17,18,19,20]. Aspect, color, taste and palatability were pleasant and the two mixtures were indistinguishable. Patients from both groups underwent four-hour hemodialysis sessions three times weekly using the same method: six patients from each group were treated with high efficiency hemodialysis and five with post-dilution hemodiafiltration with ultrafiltration, corresponding to 40% of dry body weight; high biocompatibility poly-ether-sulfone membranes were used to ensure dialysis adequacy with an equilibrated Kt/V [21] (eqKt/v) ≥1.2 in all patients. Patients were instructed to consume a standard breakfast (200 Kcal and 8–10 g protein) at least 90 min prior to treatment. Both groups underwent the following tests at the start and after 6 months of amino acid or placebo: sampling of arterial blood from arteriovenous fistula established prior to onset of hemodialysis session.

### 2.1. Hemodialys Adequacy Measurement

Mean weight at the start and end of dialysis, measurement of inter-dialytic weight gain, calculation of eqKt/V and Equilibrated Protein Catabolic Rate (ePCR), taking into account times (40–60 min after end of HD) of exhaustion of the compartmental imbalance rebound disequilibrium [22].

### 2.2. Blood Chemistry Checks

Measurement of blood urea nitrogen, creatinine, serum calcium, phosphoremia, total protein and albumin ratio, hemoglobin, total, HDL and LDL cholesterol, triglycerides, glycemia, sodium, potassium, PTH, C-reactive protein, alkaline phosphatase, leukocyte and lymphocyte count, IgG, IgA, IgM, C3, C4, serum iron, transferrin, ferritin, pH, bicarbonatemia.

### 2.3. Pharmacological Monitoring

Mean dose of oral or i.v. iron administered over the previous month, mean weekly dosage of erythropoietin, erythropoietin resistance index (ERI), administration of anti-hypertensive drugs, vitamin D derivatives, calcimimetics and oral antilipemic agents.

### 2.4. Monitoring of CLINICAL Outcome

Morbidity, days of hospitalization, mortality, death and cause of death, other causes of drop-out blood pressure monitoring pre- and post-dialysis.

### 2.5. Echocardiographic Evaluations

During the mid-week inter-dialytic period a color doppler M-mode echocardiogram (Ultrasound EsaOte MyLabXPro30—Genova, Italy) and 1D/2D measurement of cardiac chambers, left ventricular mass and ejection fraction was performed as recommended by the American Society of Echocardiography [23] with an assessment of Left Ventricular Ejection Fraction; LVIDd: Left Ventricular Index Diastolic Diameter; IVS: Interventricular septum at End Systole; FCS: Fraction Cardiac Shortening; LVMI: Left Ventricular Mass Index; Peak E: Peak Early Diastolic Filling; Peak A: Peak Late diastolic filling; E/A ratio: Ratio between Peak E and Peak A; LW: Lateral Wall; Peak Em: Myorcadial proto-diastolic Speed Peak; Peak Am: Atrial Myorcadial Diastolic Speed Peak; Em/Am Ratio: Lateral Wall Ratio between Em and Am; LW IVRTm: Left Wall Isovolumetric myocardial Relaxation Time; IVS Sm: Interventricular Septum Myocardial Systolic Speed; IVS Peak Em: Interventricular Septum Peak Em; IVS: Interventricular Septum Peak Am; IVS IVRT: Isovolumetric relaxation time of the interventricular septum. All cardiac chamber measurements (end-diastolic dimensions, end-systolic dimensions, thickness of the interventricular septum, thickness of the posterior wall and derived measures such as mass and ejection fraction) were obtained in an Apical 4-chamber. According to the protocol, tissue doppler tracings were obtained for the interventricular septum and the lateral wall of the left ventricle in the Apical 4-chamber. Bioelectrical impedance analysis (BIA) was performed using a Renal EFG50 KHz; EFG Diagnostic Ltd., (Belfast, Northern Ireland) to measure height, body surface area, reactance (Rz), resistance (Xc), and subsequently calculate: phase angle (PhA), Total Body Water (TBW), Extracellular Body Water (ECW), Intracellular Body Water (IBW), Free Fat Mass (FFM), Body Cellular Mass (BCM), muscle mass (MM), Fat Mass (FM), BMI and Energy Expenditure. BIA testing was carried out 40–60 min after the end of hemodialysis treatment. A blood sample for monitoring of the AAs profile was collected through the arteriovenous fistula (arterial blood) prior to connecting the patient to the hemodialysis monitor and the following were measured: Asparagine, Glutamic Acid, Aspartate, Serine, Glutamine, Histidine, Glycine, Threonine, Alanine, Arginine, Tyrosine, Cysteine, Valine, Methionine, Tryptophan, Phenylalanine, Leucine, Isoleucine, Lysine, Proline.

The study was conducted in accordance with the Declaration of Helsinki, and the protocol was approved by the Ethics Committee of ASL Cagliari (nr.135/2019/CE 02-12-2019)

### 2.6. Amino Acid Identification Methods 

To ensure the highest degree of reliability, accurate methods were applied in the sampling and collection of arterial blood for use in AA concentration assays before and after the dialysis session. 10 mL of whole blood were collected in 2 heparinized test tubes and stored at room temperature (to avoid issues of thermal hydrolysis). Plasma was separated within 2 h of collection by centrifuging at 3000 rpm for 10 min. Plasma thus obtained was mixed to obtain a homogenous solution; 2-mL samples were then stored in cryogenic test tubes at a temperature of 220 Celsius Degree. Within 2 days of collection, samples were transferred on dry ice to the laboratory for final storage. The analytical method required ‘‘pre-column’’ derivation of free amino acids by ortho-phthalaldehyde and 9-fluorenyl-methyl-chloroformate for the recognition of primary and secondary amino acids, respectively. Derivatives were separated by means of reverse-phase liquid chromatography and revealed using a fluorometer X-LC (model 3020FP). Analysis was carried out on a 1 mL sample of a standard mixture or serum. Sample testing was preceded by the analysis of a standard mixture to verify system efficiency. Graduated concentrations (from 29 to 233 mM/mL) of the standard mixture were used to establish calibration curve for subsequent use in quantitative analysis. To boost reliability of results, each sample was analyzed in triplicate and each amino acid quantified based on the mean obtained from three determinations. Results were obtained by injecting 1 mL of mixture and simultaneously measuring absorbance at 338 nm and 262 nm. Samples were tested using an amino acid HPLC X-LC-Jasco analyzer linked to an HP ProDesk elaborator. AA concentrations were expressed in both ml/L and mg/dL and compared with standard values in our laboratory. Twenty AAs were determined, including Total AAs (TAAs), Essential AAs (EAAs) including Branched-Chain AAs (BCAAs) and Non-Essential AAs (NEAAs).

### 2.7. Statistical Analysis

Central tendency and the dispersion of continuous variables are reported as mean ± standard deviation. Descriptive statistics for categorical variables are reported as number and percentage frequency. Due to violations of the normality assumption (Shapiro–Wilk test), hypothesis testing was based on non-parametric statistics. Within- and between-group comparisons for continuous variables were carried out using the Wilcoxon signed rank test and the Mann–Whitney U-test, respectively. Dichotomous variables were compared using the Chi-square test or Fisher exact test, if appropriate. Values at T0 were compared with values at T6 (within group) to assess the presence of a time effect. The effect of treatment over time (interaction time x treatment) was investigated by computing the difference (δ) of values at T6—values at T0 and comparing these deltas between the two groups. All tests were two-tailed. A *p* value of <0.05 was considered statistically significant. All statistical analyses were carried out using the SAS/STAT statistical package, release 9.4 (SAS Institute Inc., Cary, NC, USA).

## 3. Results

*Amino acid profile*: Comparison of changes in levels of aminoacidemia between the placebo group and the group treated with the novel amino acid mixture is illustrated in Table 1.

Levels of thirteen AAs remained unchanged from the start of the study up to the 6th month of observation. Both variations over time within each group and the change in delta (δ) values between the two groups were analyzed at the start of the study and after a period of 6 months. Glutamic acid displayed a significant decrease over time only in the TRG group, although a significant difference was recorded in δ values versus PLG. Asparagine and histidine levels decreased considerably in the two groups, although δ values over time between the two groups did not differ significantly. Alanine levels fell significantly in PLG, whilst remaining constant in TRG, displaying a change in δ values over time between the two groups. Tryptophan decreased in PLG, but increased significantly in TRG, with a significant inter-group difference in δ values. No other significant differences were observed for the remaining 14 AAs. Table 2 illustrates blood chemistry tests performed at the start of the study and after a treatment period of 6 months.

*Nutritional parameters*: Equilibrated Protein Catabolic Rate, used to determine protein uptake in g/kg, takes into account plasma rebound of urea released from body fluid compartments that commences on the termination of the hemodialysis session [24]. The data highlighted a stable protein intake of approx. 1.0 g/Kg/day in all patients from both groups with an indicative caloric value provided by Resting Energy Expenditure of between 20 and 22 Kcal/kg/day (Table 3).

No differences in nutritional indices (Body Mass Index, total protein, albuminemia, complement, lymphocytes, transferrin, cholesterol) were detected between the two groups. It is interesting to observe how better control of fasting glycemia levels elicited a return to optimal levels in TRG, with PLG displaying a tendency towards poorer control. The amount of anti-lipemic agents prescribed (statins and/or ezetimibe) over the 6-month study period corresponded to 10.0 ± 7.5 mg/day in TRG and 8.2 ± 7.5 mg/day in PLG (*p* = 0.06). Post-dialysis body weight was characterized by a non-significant change in TRG. However, as illustrated in Table 3, a significant deterioration of the phase angle (PhA) and subsequent improvement in line with increase in muscle mass (<0.05) was observed. In the patient population studied, PhA values were not affected by the difference between ECW and TBW, but rather by Fat Free Mass and, in particular, by a significant decline of muscle mass in PLG compared to the increase observed in TRG.

*Bone metabolism*: No changes in PTH were observed in either of the two groups, although levels were consistently higher in the TRG group throughout the entire observation period. Nevertheless, although stable, lower levels of calcemia were detected in TRG versus PLG, with levels falling significantly in PLG over a 6-month period; although not statistically significant, phosphorus was found to be well controlled at optimal levels and alkaline phosphatase remained consistently within normal range in both groups. As higher PTH levels were observed in TRG from study outset, weekly mean doses of drugs used to treat secondary hyperparathyroidism were adjusted; calcitriol/paricalcitol: TRG start 6.8 ± 9.04 µg/week versus 3.02 ± 3.2 µg/week after six months (*p* = 0.03); PLG start 6.5 ± 8.2 µg/week versus 5.2 ± 6.3 µg/week after 6 months (*p*: n.s.); cholecalciferol: TRG start 7.954 ± 6.306 UI/week versus UI/week 14.069 ± 7.101 after six months (*p* = 0.009); PLG start 9.309 ± 7.584 UI/week versus 13.068 ± 9.037 UI/week after 6 months (*p* < 0.05), use of calcimimetics was limited to etelcalcetide: TRG start 4.7 ± 2.7 mg/week versus 2.7 ± 6.0 mg/week after 6 months (*p* < 0.05); PLG start 2.7 ± 6.0 mg/week versus 2.7 ± 6.0 after 6 months (*p*: n.s.). However, no significant differences were detected in the change to δ values between the two groups.

*Hematological findings*: Table 4 shows how, despite the lack of significant changes to Hb, the iron pool and the dosage of orally and/or parenterally-administered iron elicited an important decrease in the need for erythropoietin to maintain the Hb values between 11 and 12 g/dL; the erythropoietin resistance index (ERI) confirms this finding.

No significant variation was observed in the number of leukocytes and lymphocytes. Transferrin was unaffected by changes to the iron pool induced by the administration of different doses of iron over the 6-month period, and likewise ferritin was not affected by the presence of potential inflammatory stimuli, as shown by a marked, although not significant, decrease in C-reactive protein levels in TRG.

*Cardiovascular system*: lastly, Table 5 shows an improvement in left ventricular ejection (LEVF) in the group treated with the novel amino acid mixture, and a significant variation over time between TRG and PLG.

Interventricular septum (IVS) showed a tendential downward-trend in the TRG group, with Peak E denoting a slight decrease in PLG and increase in the treated group. TRG displayed a tendency towards increased diastolic volume. Lastly, systolic myocardial velocity in the interventricular septum performed better in ventricular filling. These parameters were unaffected by changes in arterial blood pressure, which was well controlled and did not differ significantly between the two groups: start values of 141.9 ± 18.6 and at six months of 144.1 ± 19.8 were recorded for systolic arterial blood pressure in PLG, whilst TRG registered start values of 138.6 ± 21.6 and 130.6 ± 20.2 at six months. Start values of 78.2 ± 11.6 and 76.5 ± 13.7 at six months were recorded for diastolic arterial blood pressure in PLG, whilst TRG registered start values of 79.5 ± 9.9 and 78.6 ± 7.8 at six months. Arterial blood pressure is reported as the mean of three measurements taken during thrice-weekly sessions. A series of anti-hypertensive drugs were prescribed (ACE inhibitors, sartans, β-blockers, α-lytics) with no significant differences in mean dosage between the two groups.

*Symptomatology, morbidity, mortality and hospitalization*: no significant differences were observed between the two groups in days of hospitalization over the 6-month observation period (TRG: 0 days, PLG: 0.54 days).

*Significant symptomatology*: Each group included one patient with atrial fibrillation. No deaths occurred in the two groups throughout the period of observation.

## 4. Discussion

In hemodialysis patients, amino acid losses via diffusion across hemodialysis membranes and the low molecular weight of AAs, ranging from a minimum of 89.1 to a maximum of 240 g/mol [9,10], similar to a wide range of low molecular weight uremic molecules [25], represent a well-known determinant factor. Moreover, these losses are further aggravated by the negative convective force of dialysate using high-flux hemodiafiltration [11], with up to 60% of body water being replaced by a poly-electrolyte-based solution used as dialysate for infusion. Additionally, in our study, arterial blood samples obtained via patients’ arteriovenous fistula were used to estimate AA concentrations, yielding significantly different findings to those obtained in other studies conducted using venous blood samples. AA plasma concentrations obtained following 6 months treatment (mixture and placebo) for aspartic acid, glutamic acid, asparagine, glutamine, alanine and serine had decreased compared to baseline levels, whilst tryptophan levels had increased in the TRG Group. Firstly, none of the amino acids listed previously were present in the amino acid mixture used, with the exception of tryptophan. Moreover, smaller decreases in lysine, threonine, phenylalanine, cysteine, valine, tryptophan and histidine concentrations were detected in TRG compared to PLG. Thanks to good patient compliance with the novel AA mixture used, a higher incorporation and anabolism of branched-chain amino acids (BCAAs) occurred, particularly on muscle energy metabolism. These changes were associated with an increase in aspartic acid, glutamic acid and asparagine which, together with BCAAs, were implicated in the production of both glutamine and alanine. This would explain, in part, the significantly higher decrease in aspartic acid, glutamic acid and asparagine and lower decrease of glutamine and alanine in TRG compared to PLG. In the Authors’ opinion, tryptophan increase was not dependent solely on supplementation provided by the AA mixture but, more likely, on a reduced degradation of tryptophan by the Kynurenine pathway [26,27]. The data obtained suggest that essential amino acids, in particular branched-chain amino acids, are capable of inducing a higher physiological production of muscle energy, enhancing gluconeogenesis and promoting a higher availability of systemic tryptophan circulation, fundamental in muscle protein synthesis and in boosting cell sensitivity to the action of insulin [28]. Hypercatabolism generated by the uremic milieu is amplified by HD; indeed, the administration of amino acid supplements may also contrast the notable hypercatabolic, inflammatory and pro-oxidant effects triggered in the hours directly following hemodialysis session [29]. Moreover, hemodialysis sessions may contribute towards limiting the effects of the well-known harmful mechanisms on gut absorption of AAs caused by catabolic derangement of the uremic microbiota [30], as well as overcoming the difficulty of AA cellular access and anabolism caused by poor sensitivity to insulin action and metabolic acidosis [31]. As shown in Table 2, potential confirmation of pro-insulin activity could be hypothesized due to the fact that administration of the novel amino acid mixture appeared to promote an improved glycemic control in TRG. One of the major metabolic deficits produced is manifested in aging patients with a long dialysis vintage and scarce physical activity at muscle level, being characterized by progressive reduction of muscle mass and force [32,33]. In our study, over a six-month period of hemodialysis treatment, a significant increase in PhA [34], indicating a change in lean mass and, in particular, muscle mass was observed in the TRG group alone. Furthermore, the Authors wish to highlight how positive effects of the novel amino acid formula are brought about in the osteo-muscular system in the TRG group despite a higher status of secondary hyperparathyroidism. Indeed, Garibotto et al. [12] previously reported that in patients affected by Stage 5 chronic kidney disease, uremia and muscle hypercatabolism, often exacerbated by hemodialysis, heavily compromised BCAAs, in particular leucine and valine, in addition to producing changes in plasma levels of arginine, tyrosine, tryptophan and cysteine. It is indeed well-known in the field of cardiology that amino acids promote the tricarboxylic acid cycle implicated in the production of energy [35], of which myocytes are avid consumers. Table 5 shows how left ventricular ejection fraction is not only increased in TRG, but also displays a marked change in δ values, whilst values remain unchanged in PLG. In the Authors’ opinion, this finding is of considerable importance when taking into consideration the shortness of restoration time for improvement in intracardiac flow and myocardial mass in hemodialysis patients. Moreover, in addition to the significant increase in ejection fraction, a shift was observed in TRG towards reduction of the interventricular septum and myocardial mass, displaying an initial improvement which, although not significant, tends to validate the finding of improved compliance and performance of the myocardial muscle. Initial changes to parameters of diastolic function (Peak E and Peak A) likewise indicate an improvement in TRG versus PLG. The latter are key signals detected by means of tissue doppler pointing to a clear early positive effect on the energy metabolism of myocytes following prolonged regular use of the new amino acid mixture. Several papers have reported how benefits yielded may be correlated with enhanced intra-myocyte uptake of asparagine, and therefore of aspartic acid, serine, glutamine and, in particular, tryptophan, highlighting a protective action on and improvement of function cardiac impairment [36,37].

*Secondary anemia*: the onset of anemia in uremic patients may be ascribed to the pro-oxidant and inflammatory action of CKD, inevitable periodic blood residues in the extracorporeal circuit at the end hemodialysis session, frequent gastrointestinal bleeding and, at times, to the reticuloendothelial system blockade of the iron pool. Essentially however, the main cause is represented by an insufficient renal production of erythropoietin already present in the early stages of CKD [38,39,40]. In our opinion, maintenance of target hemoglobin values with a reduced need for erythropoiesis-stimulating agents, demonstrated by a progressive 36% reduction in TRG versus 7.5% in PLG, may correlate with a positive anabolic effect produced by the administration of AAs, in particular EAAs. Similar findings were previously reported by Bolasco et al. [41] in a short-term randomized study between a group treated with a mixture of EAAs and NEAAs without metabolic accelerators such as malic acid or succinic acid, and a control group. Other researchers had previously demonstrated a stimulatory action of erythropoietin production and significant positive correlation between hemoglobin and levels of histidine and other EAAs such as tryptophan. It is an acknowledged fact that the histidine present in the new mixture improves anemia [42,43], thanks to its predominant role in hemoprotein synthesis [44].

## 5. Conclusions

The action of the novel amino acid mixture opens new horizons for the development of a replacement/integrative therapy using AAs, which may be tailored to meet the requirements of patients undergoing hemodialysis. However, one limitation to the study is the small sample size: although we adopted a randomized comparative double-blind study design, the difficulty of identifying hemodialysis patients who had not taken nutritional supplements over the previous year was an extremely difficult task, particularly as, in addition to the exclusion criteria adopted, we sought to select and recruit patients who would comply fully with taking regular amino acid supplements over a 6-month period. The effects produced by the amino acid mixture support the presence of an effective metabolic boost, producing a positive result on metabolism and cardiac performance, stabilizing hemoglobin levels, thus implying a need for lower doses of erythropoietin, increasing cell sensitivity to the key actions of insulin, and improving muscle metabolism with slower loss of volume. The study is still ongoing and the results obtained should be confirmed in future studies. However, the findings obtained to date lead us to assert that AA supplementation, specifically EAAs and BCAAs, should become an essential aspect in the care of hemodialysis patients.

## Figures and Tables

**Table 1 nutrients-14-03492-t001:** Mean plasma concentrations of amino acids in treatment and placebo groups and differences observed in temporal inter-study (δ) and intra-study group variations.

Amino Acid Plasma Concentrations (mg/dL)	Placebo Group at Start of Study	Placebo Group after 6 Months	Treatment Group at Start of Study	Treatment Group after 6 Months	Placebo Group—δ Variation over 6 Months	Treatment Group—δ Variation over 6 Months
Aspartic acid	0.37 ± 0.11 *	0.23 ± 0.13 *	0.48 ± 0.24 *	0.22 ± 0.09 *	−0.14 ± 0.10	−0.26 ± 0.20
Glutamic acid	2.19 ± 0.42	1.99 ± 0.56	2.57 ± 0.67 ^Δ^	1.72 ± 0.34 ^Δ^	−0.20 ± 0.50 ^◊^	−0.86 ± 0.85 ^◊^
Asparagine	0.75 ± 0.30 ^⊗^	0.44 ± 0.16 ^⊗^	0.87 ± 0.33 ^⊗^	0.49 ± 0.11 ^⊗^	−0.31 ± 0.36	−0.38 ± 0.35
Serine	0.69 ± 0.19	0.59 ± 0.12	0.76 ± 0.25	0.58 ± 0.15	−0.10 ± 0.16	−0.18 ± 0.30
Glutamine	7.09 ± 2.38 ^Δ^	4.85 ± 1.11^Δ^	6.15 ± 2.93	4.98 ± 1.15	−2.24 ± 2.93	−1.17 ± 3.11
Histidine	1.81 ± 0.69 ^⊗^	0.90 ± 0.16 ^⊗^	2.18 ± 1.16 ^⊗^	1.21 ± 0.49 ^⊗^	−0.91 ± 0.64	−0.97 ± 1.06
Glycine	1.53 ± 0.46	1.48 ± 0.42	1.55 ± 0.57	1.59 ± 0.52	−0.05 ± 0.65	0.05 ± 0.65
Threonine	1.46 ± 0.64	1.15 ± 0.38	1.42 ± 0.62	1.34 ± 0.47	−0.31 ± 0.88	−0.08 ± 0.59
Alanine	3.50 ± 0.70 *	2.92 ± 0.46 *	3.32 ± 0.63	3.32 ± 0.86	−0.58 ± 0.49 ^◊^	−0.00 ± 0.93 ^◊^
Arginine	3.10 ± 0.57	3.09 ± 0.83	2.81 ± 1.00	3.60 ± 1.23	−0.01 ± 1.13	0.79 ± 1.42
Tyrosine	0.55 ± 0.13	0.58 ± 0.21	0.54 ± 0.12	0.54 ± 0.11	0.03 ± 0.20	0.00 ± 0.12
Cysteine	4.91 ± 0.90	4.84 ± 0.98	5.05 ± 1.42	5.12 ± 1.55	−0.07 ± 1.19	0.07 ± 1.34
Valine	1.76 ± 0.35	2.03 ± 0.41	1.66 ± 0.43	1.78 ± 0.41	0.26 ± 0.51	0.12 ± 0.57
Methionine	0.46 ± 0.11	0.48 ± 0.22	0.49 ± 0.19	0.38 ± 0.08	0.02 ± 0.21	−0.11 ± 0.21
Tryptophan	0.91 ± 0.28	0.97 ± 0.46	0.74 ± 0.29 ^#^	0.97 ± 0.30 ^#^	0.06 ± 0.62 ^θ^	0.23 ± 0.29 ^θ^
Phenylalanine	0.94 ± 0.23	0.90 ± 0.24	0.89 ± 0.22	0.87 ± 0.29	−0.04 ± 0.26	−0.03 ± 0.33
Isoleucine	0.78 ± 0.21	0.84 ± 0.22	0.80 ± 0.16	0.70 ± 0.25	0.06 ± 0.36	−0.10 ± 0.33
Leucine	1.13 ± 0.25	1.29 ± 0.32	1.16 ± 0.29	1.05 ± 0.26	0.16 ± 0.40	−0.11 ± 0.43
Lysine	1.99 ± 0.47	2.17 ± 0.83	1.94 ± 0.53	2.10 ± 0.63	0.17 ± 0.77	0.16 ± 0.81
Proline	4.26 ± 1.32	4.22 ± 1.86	4.73 ± 1.31	4.23 ± 1.85	−0.03 ± 1.89	−0.51 ± 2.05

Inter-Study Group variation δ: ◊: *p* < 0.04; θ: *p* < 0.05; Intra-Study Group variation * *p* = 0.002; #: *p* = 0.024; *p* = 0.005; ⊗: *p* < 0.001; Δ: *p* = 0.032.

**Table 2 nutrients-14-03492-t002:** Pre-Dialysis Blood Chemistry Analysis of Treatment and Placebo Groups and Differences observed in Temporal Inter-Study (δ) and Intra-Study Group Variations.

Parameters	Placebo Group at Start of Study	Placebo Group after 6 Months	Treatment Group at Start of Study	Treatment Group after 6 Months	Placebo Group—δ Variation over 6 Months	Treatement Group—δ Variation over 6 Months
Dry weight, kg	67.85 ± 14.74	67.12 ± 14.49	62.73 ± 16.16	63.18 ± 17.39	−0.73 ± 10.43	0.45 ± 10.06
Interdialytic weight gain, kg	2.78 ± 0.94	2.66 ± 0.86	2.23 ± 0.68	2.73 ± 0.67	−0.11 ± 0.68	0.50 ± 0.89
Equilibrated Kt/V	1.33 ± 0.11	1.35 ± 0.23	1.29 ± 0.19	1.39 ± 0.22	0.01 ± 0.18	0.21 ± 0.29
Equilibrated Protein Catabolic Rate, g/kg/day	1.02 ± 0.11	0.99 ± 0.22	1.06 ± 0.02	1.03 ± 0.23	−0.03 ± 0.20	−0.03 ± 0.01
Blood Urea Nitrogen, mg/dL	70.32 ± 12.21	67.01 ± 11.93	67.87 ± 14.79	63.55 ± 19.20	−3.31 ± 16.68	−4.33 ± 17.99
Creatinine, mg/dL	9.98 ± 2.22	10.54 ± 2.24	9.26 ± 2.29	9.40 ± 2.00	0.56 ± 2.17	0.14 ± 3.21
Calcium, mg/dL	9.09 ± 0.50 ^#^	8.53 ± 0.56 ^#^	8.74 ± 0.75	8.82 ± 0.63	−0.57 ± 0.69 ^θ^	0.08 ± 0.78 ^θ^
Phosphorus, mg/dL	5.09 ± 0.70	4.77 ± 0.92	5.27 ± 1.34	4.66 ± 1.30	−0.32 ± 1.09	−0.61 ± 1.41
Albumin, g/dL	3.65 ± 0.30	3.75 ± 0.25	3.76 ± 0.28	3.71 ± 0.39	0.10 ± 0.25	−0.05 ± 0.41
Total Protein, g/dL	6.29 ± 0.23	6.26 ± 0.31	6.68 ± 0.71	6.44 ± 0.77	−0.03 ± 0.31	−0.25 ± 0.65
Hb, g/dL	11.12 ± 0.72	11.28 ± 0.73	11.49 ± 0.86	11.64 ± 0.81	0.16 ± 1.10	0.14 ± 0.68
Total Cholesterol, mg/dL	140.5 ± 34.2	147.9 ± 24.7	159.5 ± 32.7	154.9 ± 43.5	7.5 ± 27.5	−4.5 ± 27.1
HDL Cholesterol, mg/dL	37.73 ± 11.40	42.36 ± 12.40	51.09 ± 20.43	44.82 ± 15.45	4.64 ± 10.83	−6.27 ± 10.90
LDL Cholesterol, mg/dL	80.36 ± 26.27	85.45 ± 26.99	85.18 ± 24.70	84.82 ± 33.44	5.09 ± 33.75	−0.36 ± 23.68
Triglycerides, mg/dL	112.2 ± 43.8	109.4 ± 50.1	130.5 ± 49.9	125.9 ± 54.2	−2.8 ± 53.6	−4.5 ± 35.1
Glycemia, mg/dL	98.9 ± 12.8	108.3 ± 19.8	121.1 ± 37.1	96.8 ± 16.7	9.4 ± 22.9 ^Δ^	−24.3 ± 39.2 ^Δ^
Uric Acid, mg/dL	6.15 ± 1.17	6.25 ± 1.27	6.56 ± 1.26	6.50 ± 1.91	0.11 ± 1.76	−0.06 ± 2.20
Sodium, mmol/L	137.3 ± 2.3	136.0 ± 3.0	137.7 ± 2.2	137.2 ± 2.1	−1.3 ± 3.2	−0.5 ± 2.9
Potassium, mEq/L	5.25 ± 0.71	5.26 ± 0.79	5.57 ± 0.75	5.24 ± 0.58	0.02 ± 0.97	−0.33 ± 0.90
iPTH, pg/mL	476.1 ± 457.2	434.1 ± 225.4	513.9 ± 379.3	583.2 ± 570.4	−42.0 ± 504.8	69.3 ± 712.9
C Reactive Protein, mg/L	4.60 ± 2.03	4.49 ± 3.15	4.07 ± 2.99	2.81 ± 0.94	−0.11 ± 3.51	−1.27 ± 2.66
Alkaline Phosphatase, UI/L	71.56 ± 17.25	88.22 ± 30.54	86.82 ± 32.66	76.00 ± 21.61	16.67 ± 29.72	−10.82 ± 36.55
Total Immunoglobulins, mg/dL	1305 ± 339	1249 ± 362	1480 ± 374	1471 ± 522	−56 ± 199	−9 ± 462
C3, mg/dL	79.58 ± 12.76	79.41 ± 5.40	97.42 ± 20.55	92.20 ± 19.28	−0.17 ± 12.68	−5.22 ± 14.56
C4, mg/dL	22.63 ± 4.20	31.49 ± 21.94	24.87 ± 7.97	26.15 ± 4.92	8.86 ± 22.17	1.28 ± 7.20
pH	7.36 ± 0.04	7.38 ± 0.03	7.34 ± 0.03	7.35 ± 0.08	0.02 ± 0.04	0.02 ± 0.07
Bicarbonates, mEq/L	22.42 ± 1.92	23.37 ± 2.97	21.31 ± 1.80	22.26 ± 4.24	0.95 ± 2.58	0.95 ± 4.39

Intra-Study Group variation #: *p* < 0.01; Inter-Study Group variation δ: θ: *p* = 0.03; Δ: *p* = 0.016. C- Reactive Protein Normal value: <10 mg/dL.

**Table 3 nutrients-14-03492-t003:** Difference in Bioimpedance Analysis between Treatment and Placebo Groups and Differences observed in Temporal Inter-Study (δ) and Intra-Study Group Variations.

Parameters	Placebo Group at Start of the Study	Placebo Group after 6 Months	Treatment Group at Start of the Study	Treatment Group after 6 Months	Placebo Group—δ Variation over 6 Months	Treatment Group—δ Variation over 6 Months
Phase Angle, degree	4.20 ± 0.99	3.99 ± 0.98	4.36 ± 1.13	4.53 ± 1.16	−0.21 ± 0.50 ^Δ^	0.16 ± 0.36 ^Δ^
Total Body Water, %	55.88 ± 5.44	56.47 ± 5.51	54.29 ± 9.48	55.15 ± 8.41	0.59 ± 5.03	0.85 ± 6.00
Extracellular Water, %	56.30 ± 7.27	58.12 ± 7.65	53.94 ± 9.52	57.69 ± 5.80	1.82 ± 5.85	3.75 ± 11.57
Intracellular Water, %	42.23 ± 7.27	40.78 ± 8.45	45.83 ± 9.04	42.10 ± 5.78	−1.45 ± 5.46	−3.73 ± 11.03
Fat Free Mass, %	70.44 ± 7.83	71.27 ± 7.00	69.82 ± 12.85	71.28 ± 11.49	0.84 ± 6.45	1.47 ± 6.96
Body Cellular Mass, %	40.78 ± 7.66	40.24 ± 8.07	44.64 ± 9.73	39.98 ± 6.42	−0.55 ± 6.01	−4.65 ± 11.90
Muscle Mass, %	38.72 ± 7.36	37.70 ± 7.77	39.82 ± 8.68	41.25 ± 8.09	−1.02 ± 5.32 ^Δ^	1.44 ± 3.31 ^Δ^
Fat Mass, %	29.36 ± 7.72	29.05 ± 7.04	32.70 ± 8.95	28.74 ± 11.50	−0.32 ± 6.20	−3.96 ± 7.31
Resting Energy Expenditure, Kcal/kg	20.74 ± 2.93	20.25 ± 3.25	22.67 ± 2.66	22.95 ± 2.41	−0.48 ± 1.57	0.28 ± 2.90
Body Mass Index, Kg/m^2^	23.75 ± 4.74	23.98 ± 5.27	24.96 ± 6.40	24.92 ± 6.27	0.24 ± 1.22	−0.05 ± 1.12

Inter-Study Group variation δ: Δ: *p* < 0.05.

**Table 4 nutrients-14-03492-t004:** Pre-Dialysis Blood Chemistry Analysis in Treatment and Placebo Groups and Differences in Temporal Inter-Study (δ) and Intra-Study Group Variations.

Parameters	Placebo Group at Start of the Study	Placebo Group after 6 Months	Treatment Group at Start of the Study	Treatment Group after 6 Months	Placebo Group—δ Variation over 6 Months	Treatment Group—δ Variation over 6 Months
Hb, g/dL	11.12 ± 0.72	11.28 ± 0.73	11.49 ± 0.86	11.64 ± 0.81	0.14 ± 0.68	0.16 ± 1.10
Blood Iron, µg /dL	48.91 ± 19.08	57.27 ± 25.08	48.73 ± 22.85	52.27 ± 11.94	3.55 ± 26.38	8.36 ± 23.72
Ferritin, ng/dL	414.0 ± 285.7	428.8 ± 219.2	444.5 ± 180.9	469.7 ± 239.2	25.2 ± 169.3	14.8 ± 258.3
Transferrin, mg/dL	188.1 ± 66.3	176.8 ± 29.6	179.0 ± 29.9	169.3 ± 27.5	−9.7 ± 37.8	−11.3 ± 72.9
Lymphocytes, mm^3^	1518 ± 678	1380 ± 247	1053 ± 569	934 ± 607	−138 ± 644	−120 ± 522
Parenteral Iron administration (i.v), mg/week	198.9 ± 228.5	278.9 ± 204.7	194.9 ± 232.1	319.1 ± 114.8	124.2 ± 304.7	80.1 ± 193.3
Erythropoietin, U.I./week	14373 ± 7337	15459 ± 4425	13205 ± 5838 ^Δ^	8444 ± 3547 ^Δ^	+4761 ± 4169 ^♦^	−1086 ± 7112 ^♦^
ERI, EPO IU/week/Kg/g/dL	19.44 ± 9.73	20.97 ± 5.79	19.37 ± 9.69 ^#^	12.64 ± 6.96 ^#^	−6.73 ± 4.64 ^θ^	1.53 ± 7.47 ^θ^

Intra-Study Group variation: # *p*: < 0.002, Δ: < 0.003; Inter-Study Group variation δ ♦: *p* = 0.021; θ: *p* < 0.001.

**Table 5 nutrients-14-03492-t005:** Analysis of Interdialytic Echocardiographic Parameters Between Treatment and Placebo Groups and Differences in Temporal Inter-Study (δ) and Intra-Study Group Variations.

	Placebo Group at Start of the Study	Placebo Group after 6 Months	Treatment Group at Start of the Study	Treatment Group after 6 Months	Placebo Group—δ Variation over 6 Months	Treatment Group—δ Variation over 6 Months
LVEF, %	65.73 ± 5.6	65.00 ± 6.3	64.09 ± 10.6 ^◊^	67.27 ± 5.7 ^◊^	−0.73 ± 4.3 ^#^	3.18 ± 6.8 ^#^
LVIDD, mm	47.91 ± 4.2	47.55 ± 4.5	44.73 ± 4.5	46.00 ± 5.1	−0.36 ± 4.1	1.27 ± 6.7
IVS, mm	12.82 ± 1.9	12.36 ± 1.4	13.09 ± 3.5	11.77 ± 1.4	−0.45 ± 2.2	−1.32 ± 3.5
FCS, %	21.00 ± 7.9	22.64 ± 9.2	21.23 ± 13.2	23.36 ± 8.6	1.64 ± 7.2	2.14 ± 11.9
LVMi, gm^−2^	202.3 ± 56.4	203.1 ± 41.1	180.5 ± 54.8	171.5 ± 44.1	0.8 ± 41.0	−9.0 ± 69.8
E/A ratio	0.69 ± 0.1	0.73 ± 0.2	0.95 ± 0.6	0.88 ± 0.4	0.04 ± 0.2	−0.08 ± 0.3
IVRT, ms	95.45 ± 13.6	96.73 ± 20.5	94.91 ± 24.8	100.64 ± 24.4	1.27 ± 23.7	5.73 ± 34.4
LW Sm, cm s^−1^	7.98 ± 3.2	8.24 ± 2.3	7.52 ± 2.1	8.16 ± 2.4	0.25 ± 1.8	0.65 ± 1.5
LW Peak Em, cm s^−1^	8.55 ± 3.2	9.42 ± 3.4	8.32 ± 2.7	9.19 ± 3.1	0.86 ± 3.2	0.87 ± 2.4
LW Peak Am, cm s^−1^	9.81 ± 3.3	11.11 ± 4.1	8.01 ± 2.9	9.23 ± 3.9	1.30 ± 3.4	1.22 ± 2.0
LW Em/Am ratio	0.83 ± 0.2	0.87 ± 0.4	1.23 ± 0.7	1.26 ± 0.7	0.04 ± 0.5	0.02 ± 0.4
LW IVRTm, ms	84.7 ± 13.	100.9 ± 19.0	100.2 ± 27.5	96.4 ± 13.1	16.2 ± 18.3	−3.8 ± 34.6
IVS Sm cm s^−1^	7.40 ± 2.5	7.27 ± 2.2	7.28 ± 3.1	7.98 ± 2.8	−0.13 ± 1.5	0.70 ± 1.9
IVS Peak Em, cm s^−1^	6.43 ± 2.2	6.83 ± 2.2	7.03 ± 2.9	6.45 ± 2.1	0.40 ± 1.9	−0.57 ± 2.0
IVS Peak Am, cm s^−1^	9.59 ± 2.8	11.36 ± 3.1	8.40 ± 3.8	8.88 ± 3.5	1.77 ± 3.3	0.48 ± 1.9
SIV Peak Am, cm s^−1^	0.63 ± 0.1	0.64 ± 0.2	0.94 ± 0.5	0.68 ± 0.2	0.01 ± 0.3	−0.27 ± 0.5
SIV IVRT, ms	84.91 ± 24.8	101.27 ± 20.6	93.09 ± 14.6	96.36 ± 19.5	16.36 ± 29.5	3.27 ± 28.3

Intra-Study Group variation: ◊: *p* < 0.03; Inter-Study Group variation δ: #; *p* = 0.045; Legend: LFEF: Left Ventricular Ejection Fraction; LVIDd: Left Ventricular Index Diastolic Diameter; IVS: Interventricular septum at End Systole; FCS: Fraction Cardiac Shortening; LVMI: Left Ventricular Mass Index; Peak E: Peak Early Diastolic Filling; Peak A: Peak Late diastolic filling; E/A ratio: Ratio between Peak E and Peak A; LW: Lateral Wall; Peak Em: Myocardial proto-diastolic Speed Peak; Peak Am: Atrial Myocardial Diastolic Speed Peak; Em/Am Ratio: Lateral Wall Ratio between Em and Am; LW IVRTm: Left Wall Isovolumetric myocardial Relaxation Time; IVS Sm: Interventricular Septum Myocardial Systolic Speed; IVS Peak Em: Interventricular Septum Peak Em; IVS: Interventricular Septum Peak Am; SIV IVRT: Isovolumetric relaxation Time of Interventricular Septum.

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
