# Peer review of "Effects of a Novel Amino Acid Formula on Nutritional and Metabolic Status, Anemia and Myocardial Function in Thrice-Weekly Hemodialysis Patients: Results of a Six-Month Randomized Double-Blind Placebo-Controlled Pilot Study"

_nutrients, 2022, doi:10.3390/nu14173492_

Round 1
Reviewer 1 Report
This study report the data of the effect a novel AA mixture in hemodialysis patients. The study design is elegant, consisting in a 6-month, randomized, double-blind. Unfortunately recruited patients were only 11 in each arm, from a pool of 164 dialysis patients (namely 13.4%).
the AA reported favourable effects on muscle mass, left ventricular hypertrophy, resistance to erythropoietin.
They concluded that this new AA mix were effective in slowing down the hypercatabolic state induced by CKD and HD.
The study is of interest but several concerns arise
- The AA claimed that only compliant patients were selected: did the protocol include a run-in period for evaluating adherence ?
- The introduction and discussion sections are too long and several repetitions exists. Please rewrite the sections, and in particular in the latter, explanation of the effects described in the result section should be given .
- The conclusion is not supported by the results, since they did not show a reduction of an hypercatabolic status that, in addition, is not documented in the studied cohort of patients at baseline. Accordingly to the selection criteria, selected patients were in stable metabolic status and free from catabolic conditions.
- An in-deep revision of scientific-english is required
- Last but not least, a lot of upsetting errors regarding units are present in tables 2-3-4. ERI units must be reported. Are the AA sure that the plasma concentration unit of Amino acids, as reported in table 1, is mg/dl ?
Author Response
Reviewer 1
Comments and Suggestions for Authors
This study reports the data of the effect a novel AA mixture in hemodialysis patients. The study design is elegant, consisting in a 6-month, randomized, double-blind. Unfortunately recruited patients were only 11 in each arm, from a pool of 164 dialysis patients (namely 13.4%). the AA reported favorable effects on muscle mass, left ventricular hypertrophy, resistance to erythropoietin. They concluded that this new AA mix were effective in slowing down the hypercatabolic state induced by CKD and HD.
As emphasized in the conclusions, the limited number of patients is due to the fact that by today in dialysis centers very few patients do not take nutritional supports, vitamins or other food supports.
The study is of interest but several concerns arise
- The AA claimed that only compliant patients were selected: did the protocol include a run-in period for evaluating adherence?
The limited number of patients unfortunately did not allow us to carry out a run-in period.
- The introduction and discussion sections are too long and several repetitions exists. Please rewrite the sections, and in particular in the latter, explanation of the effects described in the result section should be given. The introduction and discussion have been revised, largely rewritten and shortened. Repetitions have been eliminated and conclusions are more concise.
- The conclusion is not supported by the results, since they did not show a reduction of an hypercatabolic status that, in addition, is not documented in the studied cohort of patients at baseline. Accordingly, to the selection criteria, selected patients were in stable metabolic status and free from catabolic conditions.
It is impossible to significantly reduce hypercatabolism induced during-after each hemodialysis session (156 HD per year). All hemodialysis patients undergo inevitable hypercatabolism during and several hours after the end of hemodialysis also patients at baseline and a reduced hypercatabolism can be obtained after 24 hours often before following HD, but this is not the rule. For the Nephrologist, the most important thing is that the patient is adequately purified by toxins (small, medium, big molecular weight). However, the aim of our work is not to reduce hypercatabolism but to confirm the metabolic effects we had already obtained with our previous work with administration of a much less effective mixture of amino acids (Bolasco P. Renal Failure 2011; 33 (1); 1-5). The aim of our work is to try to promote an anabolic effect on tissues with a high cellular metabolic turnover. And the results obtained are encouraging. The same parameters are well showed at baseline and during the study period: a good control of the levels of Blood Urea Nitrogen, potassium, phosphorus, pH, bicarbonates, C reactive protein and Protein Catabolic Rate etc. already indirectly define that there is no increase in hypercatabolism and this choice has rewarded us by obtaining a stable hemoglobin with a third dose of erythropoietin administered, an improvement in cardiac performance, a better cellular use of blood sugar. These results are an extraordinary result for us nephrologists in hemodialysis patients.
- An in-deep revision of scientific-english is required
the translation was carried out and reviewed by a British mother-tongue professional scientific translator from the University of Oxford.
- Last but not least, a lot of upsetting errors regarding units are present in tables 2-3-4. ERI units must be reported. Are the AA sure that the plasma concentration unit of Amino acids, as reported in table 1, is mg/dL? Thank you for your suggestion. However, to facilitate the dimensional understanding for nutritionist nephrologists (using the same measures units for evaluating clearance and dialysance) we have always used the measurement in mg / dL obtained from the rM/ mL. The errors have been corrected and the unit of measure has been added to the ERI.
Reviewer 2 Report
A 6-month pilot randomised study of nutritional suppelement in dialysis patients.
Lines 40 to 84 are out of scope of this study and should be deleted and introduction should probably be limited to lines 90-110
It is unfortunate that only 22 patients could have been included over more than 160 patients ; this considerably limits the value of the study which should be qualified as a « pilot study »
The sentence line 140-141 reads the same as line 146-147 ; maybe it could be mentioned only once ?
If i am correct, the total of aminoacids in one sachet is 4.5 gr ; this should be indicated, as well as the total amount in gr given per week, to allow for a daily estimate of the supplemental aminoacids.
Results
There are more than 100 statistical tests for only 22 patients studied. This is not relevant and severely exposes to beta-type error. Bonferroni correction or other process should be performed. I suggest that authors should limit their analysis to one or two most important domains : aminoacid metabolism, nutritional parameters, cardiac function or anemia management but not all.
It seems that with only 11 patients per group, echcardiography data are less robust that biochemical ones.
Finally, I am uneasy with the aminoacid variation from baseline to 6 months in both groups : they seem to decrease for many of them, which is not common : either in stable patients or during a nutritional intervention. The fact that plasma amino acids decrease may in fact decrease protein anabolism, a well documented fact. Maybe authors have some ideas or suggestions to bring to the readers ?
Author Response
Reviewer 2
Comments and Suggestions for Authors
A 6-month pilot randomized study of nutritional supplement in dialysis patients. The title was modified following our suggestion. Thank you.
Lines 40 to 84 are out of scope of this study and should be deleted and introduction should probably be limited to lines 90-110. We followed your advice leaving only a small premise (in revised paper lines 40 to 49) about the damages caused by advanced uremia to hemodialysis lines 90-110 in revised paper are now lines 49 to 77)
It is unfortunate that only 22 patients could have been included over more than 160 patients; this considerably limits the value of the study which should be qualified as a « pilot study » Yes of course, thank to your suggestion we have corrected the title as “pilot).
The sentence line 140-141 reads the same as line 146-147; maybe it could be mentioned only once? thanks a lot for reporting this typo.Done.
If I am correct, the total of aminoacids in one sachet is 4.5 gr; this should be indicated, as well as the total amount in gr given per week, to allow for a daily estimate of the supplemental aminoacids. Infact, we have added in the results the total weekly amount of amino acids = 31.5 g.
Results
There are more than 100 statistical tests for only 22 patients studied. This is not relevant and severely exposes to beta-type error. Bonferroni correction or other process should be performed. I suggest that authors should limit their analysis to one or two most important domains: aminoacid metabolism, nutritional parameters, cardiac function or anemia management but not all. We did not use Bonferroni correction to defend the choice of analyzing a large number of variables. Many Nephrologists are reading Nutrients Journal and for their better understanding it is important to show the most routine parameters used in dialysis routine. Further, the second aim is to justify the choice not to apply correction criteria.
It seems that with only 11 patients per group, echocardiography data are less robust that biochemical ones. On hemodialysis patients, especially those over 65 years of age, there is a progressive loss of the Ejection Ventricular Fraction (LEVF) even in medium-short times new incident patients due to the increase in extracellular water during the interdialytic period and result in an increased cardiac overload especially in the long inter-dialytic weekly session period. This is all made worse by the high frequency of arterial hypertension in HD population. Whereas water weight gains in anuric patients are not showed difference between intra-inter-groups nor at baseline neither at the end of the study. For us nephrologists an improved of the LEVF during six months even slightly and not to have worsened the IVS peak and the diastolic parameters is an exceptional thing.
Finally, I am uneasy with the amino acid variation from baseline to 6 months in both groups: they seem to decrease for many of them, which is not common: either in stable patients or during a nutritional intervention. The fact that plasma amino acids decrease may in fact decrease protein anabolism, a well-documented fact. Maybe authors have some ideas or suggestions to bring to the readers? Of course, our belief that the importance of amino acids in dialysis is never taken seriously. Patients on hemodialysis can lose nearly 0.7 to 1.0 kg/year of amino acids from the dialysis filter. So always give amino acids orally to all patients, especially the elderly. But, in our routine if it is necessary, administering them to all hemodialysis patients (even those who still urinate) is essential. Unfortunately, in the few studies conducted on the aminoacids metabolism in hemodialysis, contradictory data are found because the clinical-pathological variables and the eating habits of patients influence both plasma levels and their cellular metabolism still largely unknown in chronic renal failure and in hemodialysis.
Lastly, I would like to clarify that the translation was carried out and reviewed by a British mother-tongue professional scientific translator from the University of Oxford.
Reviewer 3 Report
This study has undertaken by appropriate methods and showed beautiful results.
1.Pleases discuss the reason why blood urea nitrogen levels did not elevate in the treated group even though increase of nitrogen intake from amino acid formula.
2.How much was the actual energy intake of the patients?
Please consider whether adding not only amino acids but also carbohydrates at the same time will further improve the effectiveness.
Author Response
Reviewer 3
This study has undertaken by appropriate methods and showed beautiful results. Thanks a lot!
1.Pleases discuss the reason why blood urea nitrogen levels did not elevate in the treated group even though increase of nitrogen intake from amino acid formula. The results of our previous studies showed that patients on EAAs did not experience an increase in blood urea nitrogen, notwithstanding the increased nitrogen intakes from EAAs. To explain this, very probably nitrogen from supplemented EAAs replaced an equivalent amount of patient habitual diet nitrogen, providing in this way a more efficient metabolic activity of aminoacids. Moreover, the administration of amino acids increases their level in plasma water and raises the difference in concentration gradient between plasma water and dialysis fluid (dialysis fluid amino acid content = 0). The increase in the plasma water / dialysis fluid concentration gradient produces a further increase in amino acid loss. Furthermore, we have chosen to administer them away from the hypercatabolic effects of hemodialysis which further avoids the catabolization of amino acids towards nitrogen products. In hemodialysis patients it could possible alternative metabolic way towards NH3 which is very rapidly eliminated by hemodialysis. In fact, in the few studies found, administering the amino acids i.v. or orally during the dialysis session nullify further and stable gains of the amino acids administered (by a greater concentration gradient). In the literature, although studies on the kinetics of amino acids in hemodialysis are rare, it is difficult to find significantly increased differences in blood urea nitrogen levels in treated patients
2.How much was the actual energy intake of the patients?We tried to prescribe 30-35 Kcal/kg/day
Please consider whether adding not only amino acids but also carbohydrates at the same time will further improve the effectiveness. Yes of course, It is mandatory to administer an adequate caloric intake. If this is <25 KCal / Kg / day there would be a risk that amino acids are used as a source of energy with a reduction in lean and muscle mass.
Round 2
Reviewer 1 Report
- the changes made in the conclusions should be reported also in the abstract
- unfortunately in Table 2 albumin still reproted as mg/dl, similarly total protein unit is still mg/dl, as well as potassium is still reported as mg/dl
Author Response
- the changes made in the conclusions should be reported also in the abstract. Thank you. Done
- unfortunately, in Table 2 albumin still reported as mg/dl, similarly total protein unit is still mg/dl, as well as potassium is still reported as mg/dl. Sorry, I forgot to correct it, Thank you. Done.
I repeat again that the paper has been carefully checked and revised by a native English speaker with expertise in the field
Reviewer 2 Report
Authors did not respond adequately or follow my advices to my 3 main observations:
1/ There are more than 100 statistical tests for only 22 patients studied. This is not relevant and severely exposes to beta-type error. Bonferroni correction or other process should be performed. I suggest that authors should limit their analysis to one or two most important domains: aminoacid metabolism, nutritional parameters, cardiac function or anemia management but not all.
2/ It seems that with only 11 patients per group, echocardiography data are less robust that biochemical ones.
3/ Finally, I am uneasy with the amino acid variation from baseline to 6 months in both groups: they seem to decrease for many of them, which is not common: either in stable patients or during a nutritional intervention. The fact that plasma amino acids decrease may in fact decrease protein anabolism, a well-documented fact.
Author Response
1/ There are more than 100 statistical tests for only 22 patients studied. This is not relevant and severely exposes to beta-type error. Bonferroni correction or other process should be performed. I suggest that authors should limit their analysis to one or two most important domains: aminoacid metabolism, nutritional parameters, cardiac function or anemia management but not all. We obviously agree also on the issue that the number of tests carried out is quite large, with an increased likelihood of Type I error (the mistaken rejection of an actually true null hypothesis). However, as far as the suggestion to limit the analysis to one or two domains is concerned, we think that, due to the exploratory nature of this investigation, considering variables spanning different domains, is of the utmost importance to provide an exhaustive view of the effects of supplementation of a novel micronutrient-enriched amino acid mixture. We know that this exposes to the risk that some significant results are obtained by chance, nevertheless, we were controlling the comparison-wise error rate, not the experiment-wise error rate. Indeed, we were not testing that the two groups are equal on ALL considered variables (the universal null hypothesis), we wanted instead to assess each variable in its own right, testing the (pre-specified) hypothesis that there was no difference between EACH specific item. Type I errors cannot decrease (the whole point of Bonferroni adjustments) without inflating type II errors (the probability of accepting the null hypothesis when the alternative is true). Therefore, I'm sorry but I don't agree with you to eliminate the variables indicated in table 2 as, I would like to repeat that the work has a strong interest for nephrologists and leaving all the parameters listed in table 2 makes the special issue more usable and attractive considering the increasingly frequent nephrological readers.
2/ It seems that with only 11 patients per group, echocardiography data are less robust that biochemical ones. Dear Reviewer, I am not still clear why you consider the cardiological results to be less important than the haemato-chemical and anthropometric data. In our opinion, even having only obtained an improvement in LVEF constitutes a very important success in HD patient (moreover the elderly) that is rarely achievable with other nutritional supports. I would like to point out that the M-Mode echocardiography performed was carried out in a very accurate and indisputable manner.
3/ Finally, I am uneasy with the amino acid variation from baseline to 6 months in both groups: they seem to decrease for many of them, which is not common: either in stable patients or during a nutritional intervention. The fact that plasma amino acids decrease may in fact decrease protein anabolism, a well-documented fact. The fate and use of amino acids in hemodialysis is not yet definitively and significantly established. Each amino acid follows a different metabolic path and today it is not possible to establish because and how much some AAs can be significantly used for the cellular metabolic level and other can remain in circulation with large negative and positive variations in their plasmatic levels. If you reread carefully, you will notice that aspartic acid, glutamic acid, asparagine, glutamine, and serine had decreased but are not present in the new mixture administered.Again, I reiterate the lack of reliable data in the literature. The results, however, do not demonstrate a "decrease protein anabolism" probably the opposite since, if this were not the case, we would not have a greater cardiac performance, an improvement in the phase angle, an improvement in sensitivity to erythropoietin thanks to the metabolic boost of histidine and other EAAs such as tryptophan to maintain desired levels of Hb. Lastly, I would like to remind you that the levels of the AAs studied are in arterial blood. AAs levels in literature studies are almost referred to venous blood plasma levels. Sorry, but no definitive literature data is present to support your claim.
I repeat again that the paper has been carefully checked and revised by a native English speaker with expertise in the field.